# The Diagnostic Significance of PDGF, EphA7, CCR5, and CCL5 Levels in Colorectal Cancer

**DOI:** 10.3390/biom9090464

**Published:** 2019-09-09

**Authors:** Muhammed Üçüncü, Murat Serilmez, Murat Sarı, Süleyman Bademler, Senem Karabulut

**Affiliations:** 1Health Science Institute, Istanbul Gelisim University, Istanbul 34310, Turkey; 2Department of Basic Oncology, Institute of Oncology, Istanbul University, Istanbul 34093, Turkey; 3Department of Medical Oncology, Institute of Oncology, Istanbul University, Istanbul 34093, Turkey; 4Department of Surgery, Institute of Oncology, Istanbul University, Istanbul 34093, Turkey

**Keywords:** CCL5, CCR5, chemokine, colorectal carcinoma, cytokine, EphA7, PDGF

## Abstract

In this study, we compared the levels of C-C chemokine receptor type 5 (CCR5), C-C motif chemokine ligand 5 (CCL5), platelet-derived growth factor (PDGF), and EphrinA7 (EphA7) in patients with colorectal carcinoma and healthy controls in order to investigate the significance and usability of these potential biomarkers in early diagnosis of colorectal cancer. The study included 70 colorectal carcinoma patients and 40 healthy individuals. The CCR5, CCL5, PDGF, and EphA7 levels were measured using ELISA in blood samples. PDGF-BB, EphA7, CCR5, and CCL5 levels of the patients with colorectal carcinoma were significantly higher compared to the control group (*p* < 0.001 for each comparison). Our logistic regression analysis (the area under the curve was 0.958) supports the notion that PDGF-BB, EphA7, and CCL5 are potential biomarkers for the diagnosis of colon cancer. The sensitivity, specificity, and positive and negative predictive values were found to be 87.9%, 87.5%, 92.1%, and 81.4%, respectively. To our knowledge, this is the first study that investigates the relationship between colorectal carcinoma and the four biomarkers CCL5, CCR5, PDGF, and EphA7. The significantly elevated levels of all these parameters in the patient group compared to the healthy controls indicate that they can be used for the early diagnosis of colorectal carcinoma.

## 1. Introduction

Colorectal carcinoma (CRC) is the most common gastrointestinal cancers worldwide and one of the most common four cancers in both sexes in Turkey. The annual number of new cases worldwide is estimated to be around 1 million, and half of these patients are reported to succumb to the disease. CRC is highly treatable when diagnosed at early stages; however, the rate of screening for CRC is still low in the general population. Studies focused on the early diagnosis of CRCs continue to assess different indicators [1,2,3,4]. The genetic defects that lead to colorectal cancer development cause the release of various molecules, inducing the growth, proliferation, and death of cancerous and normal cells [5]. Immune cells, fibroblasts, extracellular matrix components, growth factors, and cytokines/chemokines all play important roles during tumor progression and the metastasis of CRC [6,7]. Although CRC is considered a single type of tumor tissue, it actually involves a heterogeneous group of pathologies. In this regard “numerous markers can be suggested for diagnostic and therapeutic uses. However, little is known about the role of the systemic inflammatory and immune mediators in CRCs [8,9]. To date, more than one hundred cytokines/chemokines have been identified as specific proteins mediating chemotaxis between cells. Cytokines/chemokines enable the chemotaxis of leukocytes and stem cells both in the case of inflammation and during homeostasis. They also act in heparin binding, angiogenesis, leukocyte degranulation and migration, the maturation of B and T cells, and leukocyte and endothelial cell communication [10,11,12].

CCR5 is a protein on the surface of leukocytes and has immune functions as a chemokine receptor. It is expressed in a variety of cells, including T cells, macrophages, dendritic cells, and eosinophils. CCR5 expression is induced selectively during malignant transformation. It has been reported in some types of breast cancer. CCR5 binds many cytokines, including the chemotactic cytokine CCL5, also known as RANTES (regulated on activation, normal T cell expressed and secreted). CCL5 is chemotactic for T cells, eosinophils, and basophils and also involved in leukocyte migration into inflamed tissues and the proliferation and activation of natural killer cells [11,12,13,14,15,16].

Ephrin receptors are tyrosine kinases that are activated in the event of contact with ephrin-expressing cells. They are involved in intercellular signal transmission, angiogenesis, embryonal development, and neurogenesis. Ephrin receptor A7 (EphA7) is released from the cell surface and blocks oncogenic signals originating from different EpHA receptors by receptor inhibition. Previous studies showed reduced levels in lymphoma cases and reported positive implications for lymphoma treatment [17,18,19].

PDGF is released from many different cell types, including fibroblasts and chondrocytes, and plays pivotal roles in cell proliferation and transformation, chemotaxis, angiogenesis, wound healing, and the regulation of apoptosis. It is also produced by leukemic cells and tumor cells in carcinomas of the liver, colon, breast, and bladder [20,21,22,23].

The previously mentioned markers that are involved in cell growth have been suggested as potential biomarkers for the early detection of cancer, however, related studies have been limited to only those investigating certain cytokines/chemokines and several types of cancer. Here, in this study we compared the levels of CCR5, CCL5, PDGF, and EphA7 in patients with colorectal carcinoma and healthy controls in order to assess these parameters in terms of early diagnosis.

## 2. Materials and Methods

### 2.1. Data Collection

A total of 110 individuals, who visited our tertiary hospital between January 2015 and May 2018, were included. Of these, 70 individuals, who were followed up due to CRC diagnosis, were designated as the Patient Group. The remaining 40 (the Control Group) were recruited from healthy individuals with no findings of carcinoma who visited our center for routine screening and those with normal inflammatory parameters (C reactive protein, Sedimentation and leukocyte levels). Individuals with a history of abdominal surgery or other malignancies, including rectal cancer, were excluded. For the measurement of the mentioned biomarkers, blood samples were collected preoperatively from the patients with a resectable tumor. The stage-4 patients who were scheduled for oncological treatment with no surgical intervention gave blood before the first course of chemotherapy. In all patients, the histological type was adenocarcinoma.

The protocol of this prospective study was approved by the ethical review board of the Istanbul Faculty of Medicine, Istanbul University, Turkey (2019-102). Written informed consent was obtained from all participants.

### 2.2. Determination of Serum Levels by ELISA

CCR5, CCL5, PDGF, and EphA7 levels in the blood samples were measured by ELISA kit (Sunred Biological Technology Co Ltd., Shanghai China) following the manufacturer’s instructions. Briefly, 40 µL of serum sample and 10 µL of CCR5 antibody were placed in the antibody-coated wells by using an automatic pipette, while a volume of 50 µL of standard preparation was placed into the other wells. Then, 50 µL of streptavidin-HRP was added to each microwell. For the formation of the antigen–antibody complex, the wells were incubated for 60 min at 37 °C. Following a washing with 300 µL of washing liquid five times and drying afterward, 50 µL of chromogen reagent A and then 50 µL of chromogen reagent B were added before incubation for 10 min at 37 °C. The color reaction was stopped by adding 50 µL of stopping solution. The absorbance and concentration values of the samples were measured by the ELISA reader (ChroMate 4300 Microplate Reader, Palm City, FL, USA) at 450 nm. The concentrations were calculated based on the standard curve.

### 2.3. Statistical Analysis

All statistical analyses were performed using SPSS v. 22 (IBM SPSS, Chicago, IL, USA). Descriptive data were expressed in numbers and percentage. Intergroup comparisons for categorical variables were conducted using Pearson’s chi-squared test and Fisher’s exact test. Continuous variables were investigated for normal distribution by the Kolmogorov–Smirnov test. Student’s *t*-test and variance analysis were used for intergroup comparison of continuous variables and comparison of mean values between multiple groups, respectively. Logistic regression (univariate and multivariate) and ROC (reciever operator characteristics curve) analyses were performed to assess the diagnostic usability of the parameters. The results were evaluated at a 95% confidence interval. In this study, *p* < 0.05 was considered statistically significant.

## 3. Results

Of the 70 patients, 46 (65.7%) were male and 24 (34.3%) were female. Of the 40 controls, 22 (55%) were male and 18 (45%) were female. Twenty-five patients were under the age of 50, while 45 of the patients were aged ≥50 years, with the median age being 56 (19–83) years and 52 (38–74) years for the patient and control groups, respectively. The difference between the two groups in terms of sex and age was not significant (*p* > 0.05 for both comparisons).

The patient group had statistically significantly higher mean levels of PDGF-BB (375 vs. 16.7 ng/L), EphA7 (31.9 vs. 1.5 ng/mL), CCR5 (123.9 vs. 24.2 pg/mL), and CCL5 (108.5 vs. 40 ng/L) compared to the control group (*p* < 0.001 for each comparison) (Table 1).

While all of the four markers were found to be effective according to univariate logistic regression analysis, only PDGF-BB, EphA7, and CCL5 were effective according to multivariate analysis, with PDGF-BB being the most effective one according to (forward) logistic regression analysis (Table 2).

While the area under the curve(AUC) was 0.894 in the PDGF-BB-based model, we determined that the patients could be diagnosed with 95.8% probability (CI: 92.6–99%) when EphA7 and CCL5 were included in addition to PDGF-BB (Table 3) (Figure 1).

In this model, the sensitivity, specificity, and positive and negative predictive values were found to be 87.9%, 87.5%, 92.1%, and 81.4%, respectively. With PDGF-BB levels above 350 ng/L or EphA7 levels above 200 ng/mL, the cancer probability reached 100% (Figure 2, Figure 3, Figure 4 and Figure 5).

## 4. Discussion

Early diagnosis is a crucial factor for reduced mortality in cancer. As such, numerous biomarkers have been investigated for the diagnosis of CRC. Based on the markedly increased levels of certain chemokines reported in certain carcinomas, recent discussion has focused on the predictive usability of these proteins in other types of cancer as well [1,3,12]. This is the first study that has investigated whether the cytokines/chemokines CCL5, CCR5, PDGF, and EphA7 have significant value for early diagnosis of colorectal carcinomas.

It has been shown in various cancer types that the CCL5/CCR5 chemokine/receptor axis is hyperactivated and involved in multiple phases of carcinogenesis, including proliferation, migration, invasion, angiogenesis, and metastatic colonization [13,14,15,16]. Moran et al. [13] suggested that CCL5 could be utilized as a biomarker and prognosticator in developing therapeutic strategies against cancer. CCL5/CCR5 was also demonstrated to be linked with inflammatory bowel disease by Ye et al. [24] and Mencarelli et al. [25] and breast cancer by Velasco-Velázquez et al. [26]. Gastric cancer cells exposed to CCL5 were shown to induce apoptosis of CD8+ T cells [15,27,28] and infiltration of T-regulatory cells [15] as an immune escape mechanism. This was stated to indicate that the tumor cells could secrete CCL5 to enhance the T-regulatory cell suppressive function in tumor microenvironment through CCL5/CCR5 signaling and facilitate the immune evasion [15].

There have been many studies investigating the link between CCL5/CCR5 and CRCs as well as other cancers [16,28,29]. Aldinucci et al. [14] reported a connection between CRC development and receptors such as CCL5 and overexpression of CCR5 has been revealed by biopsies in CRCs [30]. It has also been reported that interfering with CCL5 signaling could be an alternative approach in the prevention of CRCs and that CCL5 blockade could be employed in chemotherapy [29]. Cambien et al. [29] showed the effects of CCL5 in CRC, demonstrating that CCL5 neutralization in mice with colon tumors implanted subcutaneously or under the liver capsule hampered the tumor development and resulted in decreased peritoneal carcinosis. They also observed that the metastatic liver resection pieces presented the highest levels of expression for CCL5 and its receptors CCR1 and CCR5 [29]. Additionally, Chen et al. [16] associated mesenchymal stem cells with CRC development by reporting a different mechanism involving CCL5. Sasaki et al. [31] demonstrated through their experimental model that CCL5 and CCR5 antagonism by gene transduction led to a decrease in the number of CRC tumor cells. Tanabe et al. [28] also showed that CCR blockade hampered the growth of colon cancer cells by hindering fibroblast accumulation. In this study, Tanabe et al. stated that CCR5 receptor blockade can be used in the treatment of colon cancer. This supports the effectiveness of CCR5 in the development of colon cancer. This suggests that CCR5 may be a marker that can be used in the early diagnosis of colon cancer.

In our study, both CCL5 (108.5 vs. 40 ng/L) and CCR5 (123.9 vs. 24.2 pg/mL) mean levels were statistically significantly higher in the patient group compared to the control group (*p* < 0.001 for both comparisons). This finding is consistent with the previously reported data indicating a link between the CCL5/CCR5 signal axis and CRC development. The finding supports the suggestion that CCL5 and CCR5 levels can be used clinically for early CRC detection.

Ephrin receptors have been reported to form the largest subgroup of the receptor tyrosine kinases family and include many oncogenes and proto-oncogenes that are effective in cell proliferation, differentiation, migration, and metastasis [17,18,19]. EphA7 is part of this family, but there have been only a limited number of studies that investigate its connection with cancer [17,18]. Wang et al. [17] observed that the EphA7 genes were overexpressed in gastric carcinoma cells in correlation with age, tumor stage, and extent of metastasis, and stated that EphA7 might play a role in gastric cancer pathogenesis and development. However, they also published another study [18] where they reported no expression of the EphA7 gene in CRC. They attributed this finding to the loss of expression in certain genes due to various genetic and epigenetic factors. Herath et al. [19] found that expression of the EphA7 gene was decreased in CRC and similarly explained it by epigenetic factors. In contrast to these data, we found that our patients with CRC had a significantly higher mean level of EphA7 protein compared to the healthy control group (31.9 vs. 1.5 ng/mL; *p* < 0.001). We think that the inconsistency between our study and the previously reported data might have been caused by methodological differences, given that the cited investigations were based on measuring the EphA7 gene expression levels, while our study measured EphA7 protein levels. However, there is still a need for further research for better clarification.

It has been suggested that the PDGF signaling pathway is effectively involved in cancer pathogenesis by partaking in the regulation of several autocrine and paracrine processes such as tumor growth, metastasis, and angiogenesis [20,32]. Further research is needed in order to determine the cancer types for which PDGF levels might have a predictive value. Tudoran et al. [32] reported markedly increased levels of PDGF in cases of cervical cancer. Farooqi et al. [22] showed that in many cancers, including CRC, the PDGF family genes were expressed at varying levels, depending on reasons like mutation and deletion. Manzat Saplacan et al. [23] also associated PDGF with CRC. In our study, we similarly found that the patients with CRC had a significantly higher mean level of PDGF compared to the healthy controls (375 vs. 16.7 ng/L; *p* < 0.001). This finding affirms the discussed link between PDGF and CRC and indicates that it can be used diagnostically.

In contrast to the other studies, we performed logistic regression and ROC analyses in our study. The analyses showed that each of PDGF, EphA7, CCR5, and CCL5 could be used separately as an indicator in the diagnosis of CRC. However, we also observed that the results were further improved in the case of combined utilization; the rate of accurate diagnosis exceeded 95%, particularly when PDGF, EphA7, and CCL5 were used together. The diagnostic value of these indicators becomes clearer when all the obtained data are assessed concurrently.

Our study has several limitations. We were not able to recruit a large number of subjects because the study was conducted within one medical center. Accordingly, the low number of the patients at different CRC stages did not allow for a staging analysis. The small sample size also precluded reliable results from being obtained from further analyses including other variables. Finally, as a natural result of the time constraints involved in this study, no justifiable assessment was possible concerning the investigated parameters’ predictive value for the very early diagnosis of the malignancies that the healthy individuals included may develop in the future.

## 5. Conclusions

To our knowledge, this is the first study that has investigated the possible link between CRC and PDGF, EphA7, CCR5, and CCL5. In conclusion, the PDGF, EphA7, CCR5, and CCL5 levels of the patients diagnosed with CRC were found to be significantly higher compared to the healthy controls. Also, the results obtained in the logistic regression and ROC analyses indicate that these four parameters can be used with a high accuracy in early CRC detection.

## Figures and Tables

**Figure 1 biomolecules-09-00464-f001:**
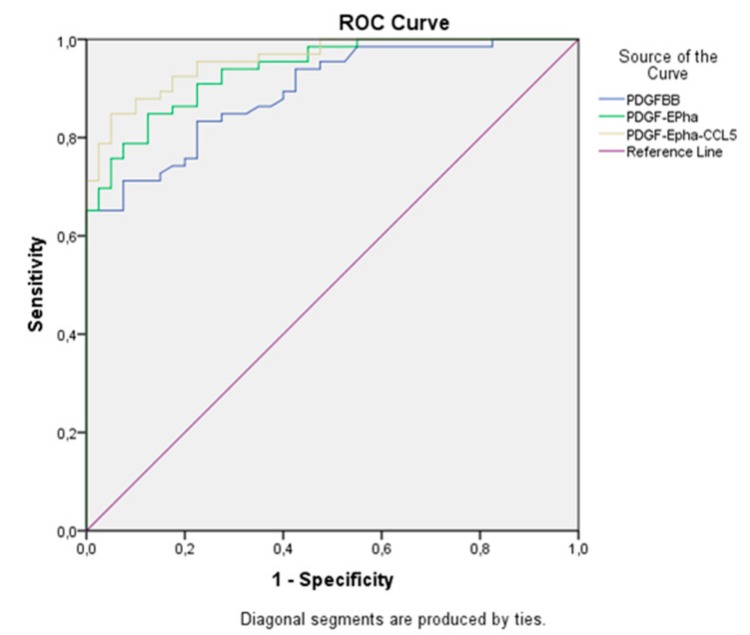
ROC curve for the combined use of markers.

**Figure 2 biomolecules-09-00464-f002:**
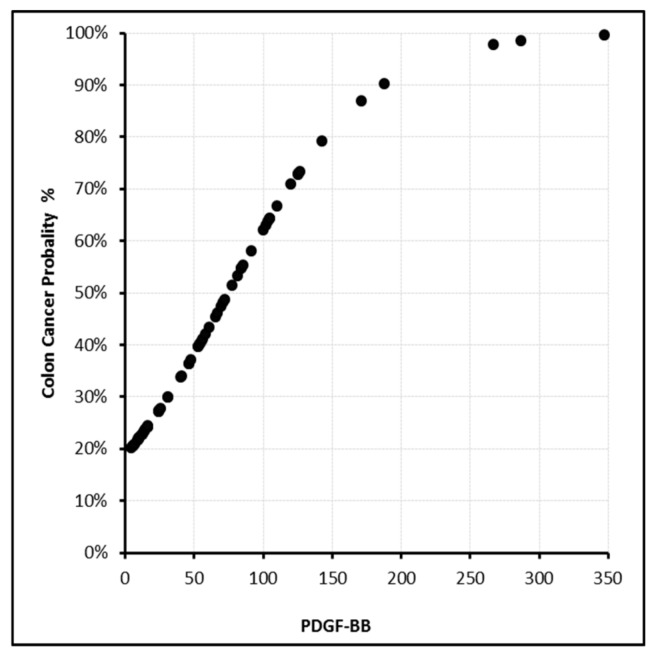
PDGF-BB (ng/L) level and cancer probability.

**Figure 3 biomolecules-09-00464-f003:**
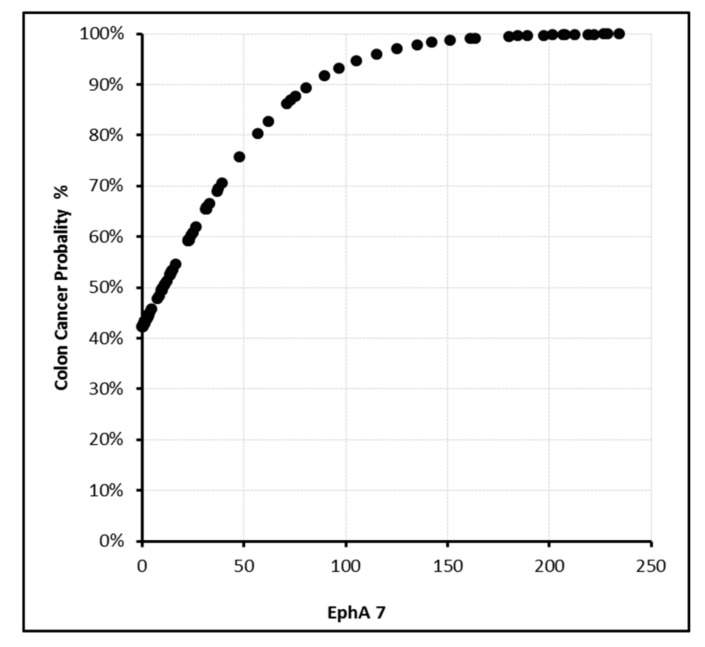
EphA7 (ng/mL) level and cancer probability.

**Figure 4 biomolecules-09-00464-f004:**
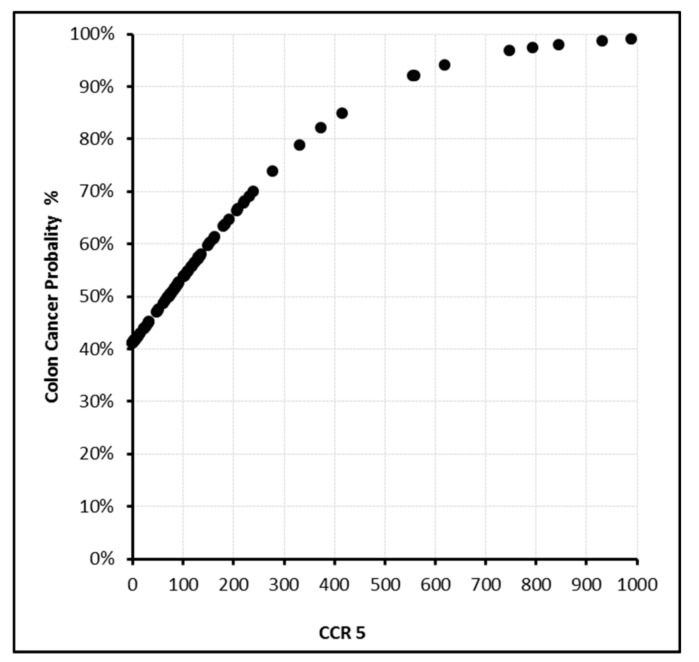
CCR5 (pg/mL) level and cancer probability.

**Figure 5 biomolecules-09-00464-f005:**
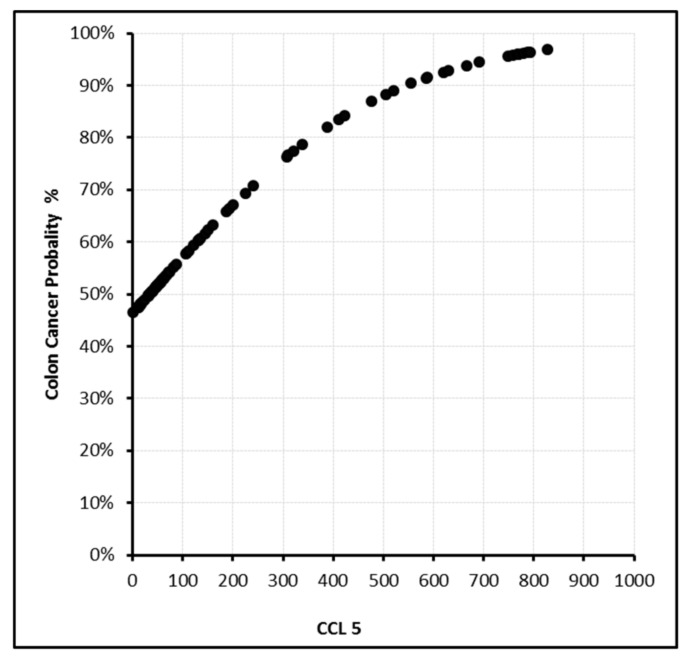
CCL5 (ng/L) level and cancer probability.

**Table 1 biomolecules-09-00464-t001:** Serum assay levels in patients with CRC and healthy controls. CRC: colorectal carcinoma; PDGF: platelet-derived growth factor; EphA7: ephrinA7; CCR5: C-C chemokine receptor type 5; CCL5: C-C motif chemokine ligand 5.

Assay	Patients (*n* = 70)		Controls (*n* = 40)		*p*
	Mean	Range	Mean	Range	
PDGF (ng/L)	375.9	9.6–4039.8	16.7	4.5–125.4	<0.001
EphA7 (ng/mL)	31.9	1.2–234.6	1.5	0.1–222.1	<0.001
CCR5 (pg/mL)	123.9	15.6–1891.5	24.2	0.4–560	<0.001
CCL5 (ng/L)	108.5	1.2–828.6	40	10.4–793	<0.001

**Table 2 biomolecules-09-00464-t002:** Univariate and multivariate analysis of the markers.

	Univariate Model					Multivariate Reduced Model				
	OR	95% CI			*p*	OR	95% CI			*p*
PDGF-BB	1.02	1.01	-	1.03	0.002	1.03	1.01	-	1.04	0.002
EphA7	1.03	1.01	-	1.05	0.003	1.02	1.01	-	1.04	0.005
CCR5	1.01	1.00	-	1.01	0.011					
CCL5	1.00	1.00	-	1.01	0.002	1.00	1.00	-	1.01	0.003
Age	1.01	0.98	-	1.05	0.245					
Sex	0.66	0.29	-	1.45	0.299					

**Table 3 biomolecules-09-00464-t003:** ROC analysis results for the combined use of markers.

Test Variables	AUC	Asymptotic Sig.^b^	Asymptotic 95% Confidence Interval	
			Lower Bound	Upper Bound
PDGF-BB	0.894	0.000	0.837	0.951
PDGF-EPhA7	0.938	0.000	0.896	0.979
PDGF-Epha-CCL5	0.958	0.000	0.926	0.990

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
