# Peer review of "The Diagnostic Significance of PDGF, EphA7, CCR5, and CCL5 Levels in Colorectal Cancer"

_biomolecules, 2019, doi:10.3390/biom9090464_

Round 1

Reviewer 1 Report

Comments to authors

My Decision is Major revisión.

Major comments

- Please, improve the discussion including the conexión between inflammation and PDGF, EphA7, CCR5,  and CCL5 levels in colon cancer progression.

-The manuscript need to be edited by a engllish native personne

-The follows published paper should be described with more detail here. Tanabe Y, Sasaki S, Mukaida N, Baba T. Blockade of the chemokine receptor, CCR5, reduces the growth of orthotopically injected colon cancer cells via limiting cancer-associated fibroblast accumulation. Oncotarget. 2016 Jul 26;7(30):48335-48345. doi: 10.18632/oncotarget.10227. This reference has been included (cyte 28) but the information is poor and must be extended in the discussion.

Minnor comments

Please, explain why there you use this desigual number of samples between cancer patients and controls (n=70 vs n=40). Is really difficult to get controls?

The authors textually indicate ¨

 To  our knowledge this is a first study to investigate the relationship between colorectal carcinoma and  the four biomarkers CCL5, CCR5, PDGF, and EphA7.

I would like to show this published paper, which is the cyted reference 28.

Tanabe Y, Sasaki S, Mukaida N, Baba T. Blockade of the chemokine receptor, CCR5, reduces the growth of orthotopically injected colon cancer cells via limiting cancer-associated fibroblast accumulation. Oncotarget. 2016 Jul 26;7(30):48335-48345. doi: 10.18632/oncotarget.10227.

We previously demonstrated that cancer-associated fibroblasts (CAFs) accumulate at tumor sites through the interaction between a chemokine, CCL3, and its receptor, CCR5, in the late phase of colitis-associated colon carcinogenesis. Here we examined the effect of a CCR5 antagonist, maraviroc, on tumor growth arising from the orthotopic injection of mouse or human colon cancer cell lines into the cecal wall by focusing on CAFs. Orthotopic injection of either cell line caused tumor formation together with leukocyte infiltration and fibroblast accumulation. Concomitant oral administration of maraviroc reduced tumor formation with few effects on leukocyte infiltration. In contrast, maraviroc reduced the intratumor number of α-smooth muscle actin-positive fibroblasts, which express epidermal growth factor, a crucial growth factor for colon cancer cell growth. These observations suggest that maraviroc or other CCR5 antagonists might act as novel anti-CRC drugs to dampen CAFs, an essential cell component for tumor progression.

These findings demonstrated that CCR5 blockade by maraviroc reduced tumor progression in colitis-associated colon carcinogénesis.  This is clear example in which CCL5 activation seem promote tumoral growth.

They also indicate

¨ The significantly elevated levels of all these parameters in the patient group compared to the healthy controls indicate that they can be used for the early diagnosis of colorectal carcinoma¨. This is true but CCR5/RANTES chemokines could be upregulated under systemic inflammatory conditions in these patients?

-How distingues these authors the relationship between chemokine expression by inflammation and/or tumoral growth chemokine-dependent levels in the present study?

-Shall you explain why elevated CCR5 /CCL5 levels promote colonorectal tumor progression by regulating PDGF, EphA7 gens in the present study?

-Are CCR5 levels induce tumoral growth or inflammation here? Please, discuss these findings from a tumoral and inmunobiological perspective in the discussion. In fact, maraviroc, a CCR5 antagonist, blocks colon tumor progression (Tanabe et al., 2016). Please, improve the discussion by adding the conexión between inflammation and PDGF, EphA7, CCR5/CCL5 levels in the context of colonrectal tumor biology.

Introd

They also indicates

 ¨However, little is known about the role of the systemic inflammatory and immune 44 mediators in CRCs [8,9]¨. Please, indicate this conexión in introduction and also discussion following my recommendation.

They also indicate ¨CCR5 expression does not occur in normal prostate and breast epithelial cells and is induced selectively during malignant transformation¨

I thing CCR5 can be also detected in normal prostate cells according this published paper. However, this CCR5 expression is low as compared to prostatic tumoral cells. This is true. Please, revise is really CCR5 is not deteced in normal prostate cells. Alpha CXCR4 and beta CCR5 chemokines can detected in ¨normal cells¨ (neurons) under migration (Merino et al., 2015, CXCR4/CXCR7 molecular involvement in neuronal and neural progenitor migration: focus in CNS repair. J Cell Physiol. 2015 Jan;230(1):27-42. doi: 10.1002/jcp.24695). In additon, chemokine blockade in¨normal cells¨ also provoke apoptosis (Merino JJ, et al. 2016. Impact of CXCR4 Blockade on the Survival of Rat Brain Cortical Neurons). Add some of these reference to demonstrate that ¨normal cells¨ also express chemokine receptors.

¨The gene expression activity of CCL2 and CCR6 was significantly higher in tumor tissue compared to adjacent normal tissue. CCL2 was also significantly higher in the blood samples of PCa patients, compared to controls. CCL5, CCL20, and CX3CL1 were lower in patient serum, compared to controls. CCR2 tissue mRNA was negatively correlated with the Gleason score and grading¨ (Tsaur et al. 2014)

Tsaur I, Noack A, Makarevic J, Oppermann E, Waaga-Gasser AM, Gasser M, Borgmann H, Huesch T, Gust KM, Reiter M, Schilling D, Bartsch G, Haferkamp A, Blaheta RA.CCL2 Chemokine as a Potential Biomarker for Prostate Cancer: A Pilot Study. Cancer Res Treat. 2015 Apr;47(2):306-12. doi: 10.4143/crt.2014.015. Epub 2014 Oct 13.

Thus, please reconsider the absence of CCR5 staining is not found in normal prostate cells.

Line 72. I am not sure if the word ¨ the usability¨ is appropiate here. If not, please, replace by another better word here.

Line 88. The described protocol for chemokine evaluation by ELISA is poor. Please, give more detail for a general audience of researchers.

Line 125. Please, give more statistical information on OR procedure used in the present study. Please, also explain the difference between univariate and multivariate models in OR analysis? This reviewer need details on described OR method in line 125.

Please, indicate the posible influence of sex or age on chemokine levels or not.

Please, also better explain the statistical methosd in line 129; Give details on how ROC was combined with other markers in the graph?. Please, include more details on OR and ROC method in the statistical section.

The fig 4 and 5 showed enhanced cancer probabilities at higher CCR5/CCL5 concentration than 250. Shall you explain this feature in the discussion? Please, indicate which units are CCR5 and CCL5 chemokines in these figures (maybe pg/ml?).

They also indicate ¨ CCL5/CCR5 was also demonstrated to be linked with inflammatory bowel disease by Ye et 156 al. [24]….. Please, explain the conexión between these beta chemokine inflammation and colonorectal cancer in the discussion.

They also indicate ¨ CCL5 neutralization in mice with colon tumors implanted subcutaneously or under the liver capsule hampered the tumor development and resulted in decreased peritoneal carcinosis¨. Please, include  published references reporting CCR5 blockade prevented tumor progression in colonorectal cancer.

I have read in the discussion this senence. ¨However, they also published another study [18] where they reported no expression of the EphA7 gene in CRC¨. How authors can explain the conexión between these beta chemokines and EphA7 gene in tumoral colon growth.

I not confident with a possible epigenic explanation for these discrepance between EphA7 and CRC genes. In fact, the conclusión indicates ¨CRC had a significantly higher mean level of EphA7 protein compared to the healthy control group¨.

Please, shall you clarify the connnexion between EphA7 and CRC5 beta chemokine receptor in the context of colonorectal tumoral growth in the present study. This part of the discussion is not clear and confuse.

Please, also explain why PDGF could increase CRC5 levels and all asscoiated signaling pathways (if is posible).

Thanks.

Author Response

First of all thank you for your evaluation. When planning this study, we planned to study these markers one by one in colon cancer patients. We found that all of these markers were significantly higher than those of healthy individuals . As a second idea, we looked at whether we could find a high probability of diagnosing colon cancer in a model that we created by combining all of these markers. And our calculations showed us that is %95,8 probabilty score.  Of course, this should be done in larger studies to be certain. We do not show that these markers increase cancer progression. İt is done already. Therefore, we do not say that the markers used here affect each other and cause cancer progression. When these markers are tested and found, the probability of CRC can be calculated. In fact, in our study, we found that there was no difference between these markers between disease stages. Logistic regression analysis already gives us the possibility of being CRC.In SPSS there is a proability score calculation part in  logistic regression . These results Show us taht the most valuable marker is PDGF (89,4%) İF we add Epha 7 and CCL5 The probality score is goes up to 95,8%. Other combinations (CCL5-CCR5, CCL5-PDGF)  were not given because the probability was lower. In word supplement we give an example

Reviewer 2 Report

The authors provide an interesting finding relating PDGF. EPhA7, CCL5 and CCR5 for biomarker discovery in colon cancer. They have shown the patients have higher levels of these proteins and thus are associated with cancer probability. Even though interesting as a preliminary clinical finding, there some concerns with the study that are as follows:

Can the author comment on whether these are diagnostic or more related with disease progression? How do the calculate disease probability should be clarified. What is the source of CCR5 in the plasma/serum? Since this is not normally secreted why do this go up in cancer? Is this release from cancer cells? Do these molecules correlate with the stage of the disease or mutational profile (if known) of at least APC, p53 etc.? Did the authors devised their own ELISA for the proteins? Please specify the catalogue no. of the Kit or the antibodies used. What does the ROC curve look like with CCR5 included? Is it possible to try other combinations like CCL5-CCR5, CCL5-PDGF etc which will give similar AUC? Units on X axis not mentioned on Figure 3-5. A description/legend with these figures will be useful.

Author Response

First of all thank you for your evaluation. When planning this study, we planned to study these markers one by one in colon cancer patients. We found that all of these markers were significantly higher than those of healthy individuals . As a second idea, we looked at whether we could find a high probability of diagnosing colon cancer in a model that we created by combining all of these markers. And our calculations showed us that is %95,8 probabilty score.  Of course, this should be done in larger studies to be certain. We do not show that these markers increase cancer progression. İt is done already. Therefore, we do not say that the markers used here affect each other and cause cancer progression. When these markers are tested and found, the probability of CRC can be calculated. In fact, in our study, we found that there was no difference between these markers between disease stages. Logistic regression analysis already gives us the possibility of being CRC.In SPSS there is a proability score calculation part in  logistic regression . These results Show us taht the most valuable marker is PDGF (89,4%) İF we add Epha 7 and CCL5 The probality score is goes up to 95,8%. Other combinations (CCL5-CCR5, CCL5-PDGF)  were not given because the probability was lower. We  can give results in example.

 We did some little changes in manuscript(in red style). We cannot find volunteers more than 40 in our study period because we sponsored by noone and we are making some other test(CRP,leukocyte levels,sedimentation, ultrasound,chest X ray and etc)  to say they are healthy.

Round 2

Reviewer 1 Report

Comments to the authors.

The manuscript has been improved now by these authors.

My Decision is accept the manuscript after including these minnor proposed changes.

I would be advisable to review the English style one more time.

I don`t need to review the manuscript again.

Minor comments

-Please, add units of ELISA in table-1 (for example if these units are ng/L, please add in the title of table-1).

Table 1. Serum assay levels in patients with CRC and healthy controls

-Please, add the symbol in lines 92-94; maybe, @ must be microliter (please put in symbol letter). Correct it.

-Please, add more details about ROC analysis in materials and methods.

Author Response

We checked the manuscript for english editing one more time and add the units. WE add the symbols.